# Subunit promotion energies for channel opening in heterotetrameric olfactory CNG channels

**Jana Schirmeyer**[1]◉, **Thomas Eick**[1]◉, **Eckhard Schulz**[2]◉, **Sabine Hummert**[1,2],
**Christian Sattler**[1], **Ralf Schmauder**[1], **Klaus Benndorf**[1]*

**1** Institute of Physiology II, Jena University Hospital, Friedrich Schiller University Jena, Jena, Germany,
**2** Schmalkalden University of Applied Sciences, Faculty of Electrical Engineering, Blechhammer, Schmalkalden, Germany

◉ These authors contributed equally to this work.
* Klaus.Benndorf@med.uni-jena.de

**Data Availability Statement:** All data are available on https://osf.io/zcuyv.

**Funding:** This work was supported by the Research Unit 2518 DynIon (project P2 to K.B.)

## Abstract

Cyclic nucleotide-gated (CNG) ion channels of olfactory sensory neurons contain three types of homologue subunits, two CNGA2 subunits, one CNGA4 subunit and one CNGB1b subunit. Each subunit carries an intracellular cyclic nucleotide binding domain (CNBD) whose occupation by up to four cyclic nucleotides evokes channel activation. Thereby, the subunits interact in a cooperative fashion. Here we studied 16 concatamers with systematically disabled, but still functional, binding sites and quantified channel activation by systems of intimately coupled state models transferred to 4D hypercubes, thereby exploiting a weak voltage dependence of the channels. We provide the complete landscape of free energies for the complex activation process of heterotetrameric channels, including 32 binding steps, in both the closed and open channel, as well as 16 closed-open isomerizations. The binding steps are specific for the subunits and show pronounced positive cooperativity for the binding of the second and the third ligand. The energetics of the closed-open isomerizations were disassembled to elementary subunit promotion energies for channel opening, $\varDelta\varDelta G_{f_{p_n}}$, adding to the free energy of the closed-open isomerization of the empty channel, $E_0$. The $\varDelta\varDelta G_{f_{p_n}}$ values are specific for the four subunits and presumably invariant for the specific patterns of liganding. In conclusion, subunit cooperativity is confined to the CNBD whereas the subunit promotion energies for channel opening are independent.

## Author summary

Olfactory sensory neurons (OSNs) in the nose transmit the information of odor molecules to electrical signals that are conducted to central parts of the brain. Olfactory cyclic nucleotide-gated (CNG) ion channels, located in the cell membrane of the OSNs, are relevant proteins in this process. These olfactory CNG channels are formed by three types of homologue subunits and each of these subunits contains a cyclic nucleotide binding domain (CNBD). A channel is activated by the binding of up to four cyclic nucleotides.

and the Collaborative Research Center Transregio 166 ReceptorLight (project A5 to K.B.) of the Deutsche Forschungsgemeinschaft. The funders had no role in study design, data collection and analysis, decision to publish, or preparation of the manuscript.

**Competing interests:** The authors have declared that no competing interests exist.

The process of channel activation is only poorly understood. Herein we analyzed this activation process in great detail by concatenating these four subunits, disabling the CNBDs by mutations and performing extended computational fit analyses providing all 32 constants for the different binding steps at different degrees of liganding and, in addition, elementary subunit promotion energies for channel opening for all subunits. Our data suggest that subunit cooperativity is confined to the action of the CNBD.

## Introduction

Cyclic nucleotide-gated (CNG) ion channels play essential roles in the signal transduction of olfactory sensory neurons and photoreceptors [1–5]. Natural olfactory cyclic nucleotide-gated (CNG) channels [2,4] are heterotetramers composed of three homologue subunits, 2×CNGA2 (A2), CNGA4 (A4), and CNGB1b (B1b) [6,7]. Each subunit contains in its intracellular C-terminus an own cyclic nucleotide-binding domain (CNBD) [2,4,8]. At heterologous expression, only the A2 subunit can form functional channels on its own [9]. In contrast, the A4 and B1b subunits introduce diverse functional effects to heteromeric channels [10–12], as modulating the sensitivity to cyclic nucleotides [7,13–15], reducing the unitary conductance compared to homomeric A2 channels [7], and mediating the action of Ca-Calmodulin [3, 7, 10–12,16–18]. Despite the fact that the A4 and the B1b subunit cannot form functional channels on their own, they also bind cyclic nucleotides and process this signal to the activation process [8,15]. The underlying complex mechanism is still elusive.

The Monod-Wyman-Changeux (MWC) model [19] has been successfully used to quantify cooperative [20] processes in proteins, thereby adopting the strongly simplifying assumptions of fixed stoichiometric factors and a joint ,allosteric' conformational change of identical subunits. Consequently, the MWC model requires at equilibrium only one constant for ligand association ($K$), one constant for the allosteric conformational change ($E0$), and one allosteric factor ($f$), i.e. the same number of constants as in a simple model containing only one binding step and two allosteric steps (Fig 1A). The MWC model, or derivatives of it, have also been repeatedly used to quantify the activation of various homomeric ion channels [21–24] including cyclic nucleotide-gated CNGA1 channels [25] and CNGA2 channels [26,27] as well as cyclic nucleotide-gated HCN2 channels [28]. To a lesser extent, the activation of heteromeric channels [29,30] has been quantified. For heteromeric channels, however, this concept must be even more vague because the different subunits require individual equilibrium constants for both ligand association ($K_x$) and the closed-open isomerization ($E_x$), resulting for the case of a heterotetrameric protein in 32 different $K_x$ and 16 different $E_x$ values (Fig 1B). To determine from experimental data such an amount of parameters seems to be completely unpromising.

In an attempt to address this question, we recently constructed a full set of 16 concatameric heteromeric CNG channels in the sequence N-A4-A2-B1b-A2-C with defined wild-type and disabled, but still functional, binding sites (Fig 1C). Concentration-activation relationships (CARs) were subjected to a global fit analysis with 16 intimately coupled state models [31]. To simplify the analyses, all $K_x$ for the two A2 subunits and all $E_x$ at equal degree of liganding were assumed to be equal. This allowed us to determine 32 $K_x$ and 4 $E_x$ with reasonable precision (S1 and S2 Tables).

However, these simplifying assumptions still bias the results because it is *a priori* neither clear that the two A2 subunits are functionally equal due to their different neighbors, as recently elegantly shown for related heteromeric CNGA1/B1 channels by structural analyses [32], nor that an equal number of ligands bound to different subunits generate an equal $E_x$.

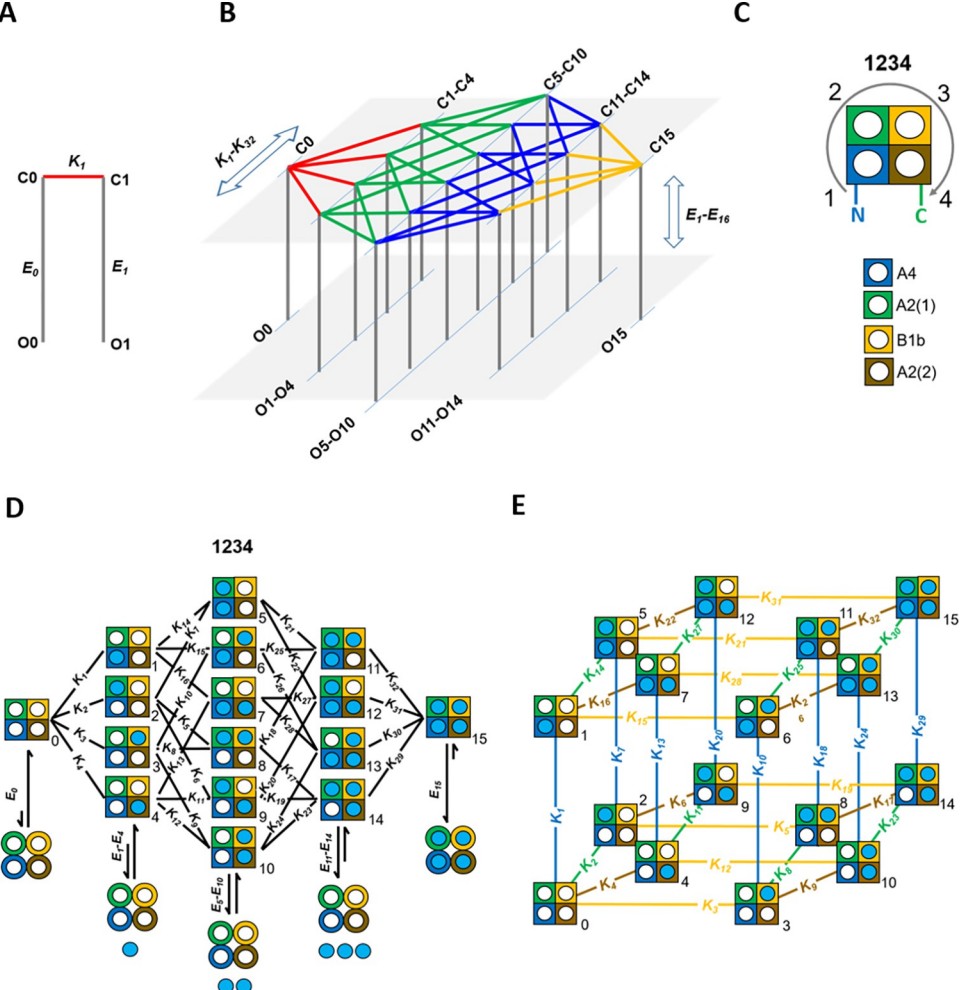

**Fig 1. State models for the activation in heteromeric proteins.** (**A**) Scheme with two closed and two open states (C0, C1, O0, O1) for a theoretical minimum ligand-gated channel. There is one ligand binding step from closed state C0 to closed state C1, specified by the equilibrium association constant $K_1$. From each closed state an opening step is possible that is specified by the respective closed-open isomerization constant $E_0$ and $E_1$. (**B**) Related scheme for a heterotetrameric channel containing 16 closed (C0-C15) and 16 open states (O0-O15), requiring for the closed channel 32 binding steps ($K_1$-$K_{32}$) and 16 allosteric opening steps ($E_0$-$E_{15}$). (**C**) Cartoon scheme of the concatamer used in the analysis with the sequence N-A4-A2(1)-B1b-A2(2)-C [31]. The colors of the subunits are used throughout the article. (**D**) Detailed cartoon scheme for the concatameric heterotrameric allosteric (HA) model used in the analysis, specifying the counting of the 32 $K_x$ and the 16 $E_x$, here summarized at equal degree of liganding. (**E**) Representation of the scheme in d in a 4D hypercube to ease the analysis.

To better understand the molecular machinery underlying the activation gating in olfactory CNG channels, our strategy of concatenation, subunit disabling, and global fitting is highly attractive because this might answer the question why these three types of homologue subunits are so different in their function. In particular, it is not clear which parts of the subunits cause these significant functional differences and which parts do interact to generate the cooperative activation and which parts do not. Furthermore, continuative analyses of our unusually consistent data set of 16 concatamers with systematically disabled subunits provides a chance to fathom in a more general way to what degree of detail such constructs can be exploited to functionally analyze heterooligomeric channels with *per se* silent subunits.

## Results

### The HACO model

We aimed to determine for heterotetrameric olfactory channels the full set of 32 equilibrium association constants, $K_x$, and 16 equilibrium closed-open isomerization constants, $E_x$. To this end we analyzed ion currents of 16 N-A4-A2-B1b-A2-C concatamers (Fig 2A and 2B) [31] differing by the number of disabled, but still functional, binding sites. In contrast to our previous report [31], we assumed here that the $E_x$ can be disassembled to elementary subunit components. An elementary subunit component is specified by the promotion factor for opening, $fp_n$ (n = 1...4), shifting the closed-open isomerization of the empty channel, specified by the constant $E_0$, more to the open state,

$$E_x = E_0 \prod_1^4 f\, p_n \tag{1}$$

For empty subunits, the $fp_n$ are unity, whereas for occupied subunits the corresponding $fp_n$ have values >1 and are specific. Hence, the 16 $E_x$ are generated by 16 combinational products of the 5 parameters $E_0, fp_1...fp_4$.

For the 16 closed states of a single heterotetrameric allosteric (HA) model (Fig 1D) and using a 4D hypercube (Fig 1E), this results in 15 independent virtual equilibrium association constants, $Z_x$ (S1 Fig), determining 32 $K_x$ [31]. The closed states of the remaining 15 models with one through four disabled binding sites were treated accordingly (S2 to S5 Figs). When using for each subunit an own disabling factor, $fd_n$ (n = 1...4), the number of free parameters for ligand binding add up to 19. Together, with the five disassembling parameters for all $E_x$, the global fit of 16 CARs requires 24 parameters. In the following, this model is termed Heteromeric Allosteric Combinational Opening ([x]HACO) model in which x specifies the number of free parameters.

We first fitted the [24]HACO model to the 16 CARs at +100 mV (S3 and S4 Tables). Compared to the simpler [17]HA model [31] (c.f. S1 and S2 Tables), the fit precision deteriorated markedly, as indicated by the relative SD (S6 Fig).

To further increase the fit constraints, we included also experimental data from 16 CARs at -100 mV, obtained from the same patches. All CARs at -100 mV were shifted to higher concentrations with respect to the data at +100 mV (Fig 2C). Similar to homomeric CNG1 and CNGA2 channels [33–35], depolarization-induced activation becomes directly evident by the current time courses when stepping from -100 mV to +100 mV (Fig 2A and 2B). In the HACO model, the voltage-dependence was assigned exclusively to the $E_x$ and not to the ligand binding at the cytosolic CNBD because it is not positioned in the transmembrane electric field. This results for the 32 CARs in the [29]HACO model containing the parameters $Z_1$-$Z_{15}$, $fd_1$-$fd_4$, $E_{0+}$, $fp_{1+}$-$fp_{4+}$, $E_{0-}$, and $fp_{1-}$-$fp_{4-}$. Signs in suffixes denote the voltages of +100 and -100 mV. The result is that the [29]HACO model is suitable to describe the gating in its intricate complexity (Fig 2C and Tables 1 and S5). Moreover, the errors of the parameters are markedly small, even smaller than with the [17]HA model (S6 Fig). When fitting the data at -100 mV with the [24]HACO model alone (S3 and S4 Tables), the errors were similarly large as at +100 mV only (S6 Fig), indicating that the superior precision of the fit with the [29]HACO model arises from the increased constraints when fitting the 32 CARs at both voltages.

### Cooperativity in ligand binding

First, the cooperativity for ligand binding at different pre-occupation of the other subunits is considered for the closed channel by both the $K_x$ values and the derived affinity increase at a

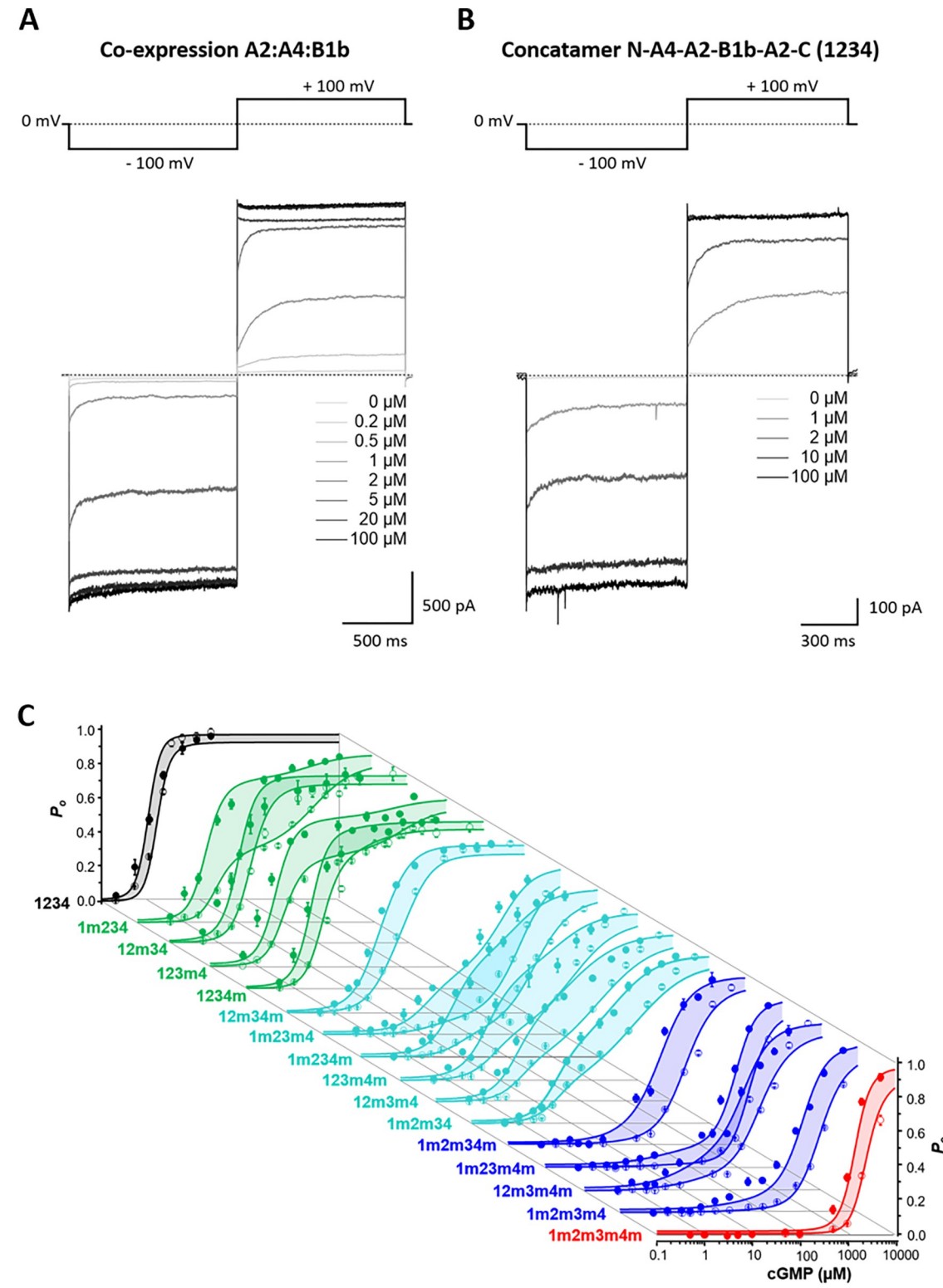

**Fig 2. Concentration-activation relationships (CARs) and cooperative affinity increase at each subunit.** Current amplitudes were measured as late currents at the end of the pulses at +100 mV and -100 mV. Stepping the voltage from negative to positive voltages generates an activating component in the time course. (**A**) Currents of heteromeric channels generated by co-expression of wt subunits with a cRNA ratio A2:A4:B1b of 2:1:1. (**B**) Currents generated by the concatamer N-A4-A2(1)-B1b-A2(2)-C show a similar activating component of the time course as co-expressed channels. (**C**) Global fit of 32 CARs with the [29]HACO model

obtained from 16 concatamers at either +100 mV (filled circles) or -100 mV (open circles). The model consists of 32 intimately coupled HA models as described in the text. For normalization of the data points see Materials and Methods. The data points at +100 mV and -100 mV were obtained from the same patches (n = 5 to 18). The curves at +100 mV and -100 mV of the same concatamer are visually related by colored areas.

subunit, $f_{ai}$ (Eq 15), relating each $K_x$ to the respective $K_x$ of the empty channel (S6A–S6D Table and Fig 3A–3D).

In the empty channel, $K_1$-$K_4$ differ only moderately by a factor of 2.7 in the sequence $K_1 > K_4 > K_2 > K_3$ (A4 > A2(2) > A2(1) > B1b) and the two A2 subunits are closely similar. Pre-occupation of one to three subunits increases the affinity at the binding sites notably, generating positive cooperativity. The degree of this increase, however, is unique for each binding site, also among the two A2 subunits.

Regarding A4 (Fig 3A), single pre-occupation of either of the three other subunits generates an $f_{ai}$ of 3.5 to 7.2, among which the two A2 neighbors exert a slightly larger effect than do the opposite B1b. When pre-occupying two subunits, a strong affinity increase on A4 is only caused by two

**Table 1. Fit parameters of the global fit with the $^{29}$HACO model at +100 mV AND -100 mV.** The dimensions of $Z_1$-$Z_4$, $Z_5$-$Z_{10}$, $Z_{11}$-$Z_{14}$, and $Z_{15}$ are $\mu M^{-1}$, $\mu M^{-2}$, $\mu M^{-3}$, and $\mu M^{-4}$, respectively. $fd_1$-$fd_4$, $E_{0+}$, $fp_{1+}$-$fp_{4+}$, $E_{0-}$, and $fp_{1-}$-$fp_{4-}$ are dimensionless.

| Specification | No. | Parameter | $^{29}$HACO model +100 mV AND -100 mV | |
|---|---|---|---|---|
| | | | value | rel. SD (%) |
| Virtual association constants | 1 | $Z_1$ | 7.31E-02 | 6.8 |
| | 2 | $Z_2$ | 3.60E-02 | 10.1 |
| | 3 | $Z_3$ | 2.66E-02 | 10.9 |
| | 4 | $Z_4$ | 4.50E-02 | 11.2 |
| | 5 | $Z_5$ | 1.14E-02 | 7.0 |
| | 6 | $Z_6$ | 6.84E-03 | 7.9 |
| | 7 | $Z_7$ | 2.37E-02 | 7.2 |
| | 8 | $Z_8$ | 7.75E-03 | 8.1 |
| | 9 | $Z_9$ | 4.35E-03 | 8.8 |
| | 10 | $Z_{10}$ | 1.90E-02 | 7.9 |
| | 11 | $Z_{11}$ | 5.18E-03 | 7.6 |
| | 12 | $Z_{12}$ | 9.98E-03 | 5.2 |
| | 13 | $Z_{13}$ | 2.41E-03 | 12.6 |
| | 14 | $Z_{14}$ | 7.54E-03 | 6.7 |
| | 15 | $Z_{15}$ | 1.80E-02 | 6.0 |
| Factors for disabling binding | 16 | $fd_1$ | 8.81E-05 | 7.5 |
| | 17 | $fd_2$ | 8.91E-03 | 8.3 |
| | 18 | $fd_3$ | 9.82E-05 | 7.5 |
| | 19 | $fd_4$ | 2.43E-03 | 9.8 |
| | 20 | $E_{0+}$ | 2.19E-02 | 3.7 |
| Factors for promoting opening at +100 mV | 21 | $fp_{1+}$ | 1.21E+01 | 3.7 |
| | 22 | $fp_{2+}$ | 6.32E+00 | 5.3 |
| | 23 | $fp_{3+}$ | 9.39E+00 | 3.6 |
| | 24 | $fp_{4+}$ | 4.18E+00 | 4.5 |
| | 25 | $E_{0-}$ | 5.93E-03 | 3.9 |
| Factors for promoting opening at -100 mV | 26 | $fp_{1-}$ | 1.74E+01 | 3.0 |
| | 27 | $fp_{2-}$ | 4.57E+00 | 3.6 |
| | 28 | $fp_{3-}$ | 7.22E+00 | 2.8 |
| | 29 | $fp_{4-}$ | 4.01E+00 | 3.2 |
| Mean s.e.m. | *** | *** | *** | 6.7 |

occupied A2 subunits, with $f_{ai}$ of 31.4. The two A2 subunits are clearly different when occupied in combination with B1b (red arrows). Pre-occupation of A2(2) and B1b together generates a four-fold lower binding affinity than pre-occupation of A2(2) alone, suggesting negative cooperativity. Triple pre-occupation does not further enhance $f_{ai}$ compared to pre-occupation of both A2 subunits. The situation of the opposite B1b subunit resembles that of A4 (Fig 3C).

Regarding A2(1) (Fig 3B), again the two neighbors A4 and B1b exert a stronger effect than the opposite A2(2) at both single pre-occupation and in combination with a pre-occupation of A2(2). In contrast to A4 and B1b, however, here triple pre-occupation causes the strongest affinity increase on A2(1). Regarding A2(2) (Fig 3D), all effects are qualitatively similar to those of A2(1) but differ quantitatively up to a factor of about 4, indicating that different neighbors of the A2 subunits exert a different function.

Together, this means for the cooperativity by binding (1) a pronounced positive and distinct negative cooperativity among the subunits, (2) stronger effects by pre-occupation of neighbor than opposite subunits, and (3) stronger positive cooperative effects of the pre-occupied A2(1) on A4 and B1b than of the pre-occupied A2(2).

## Subunit promotion energies for channel opening

To ease the considerations, all 16 $E_x$ were translated to Gibbs free energies (S7A and S7B Table) according to

$$\Delta\Delta G_{Ex} = -RT\ln\left(E_0 \prod_1^4 fp_n\right) = -RT\ln(E_0) - RT\sum_1^4 \ln(fp_n) \tag{2}$$

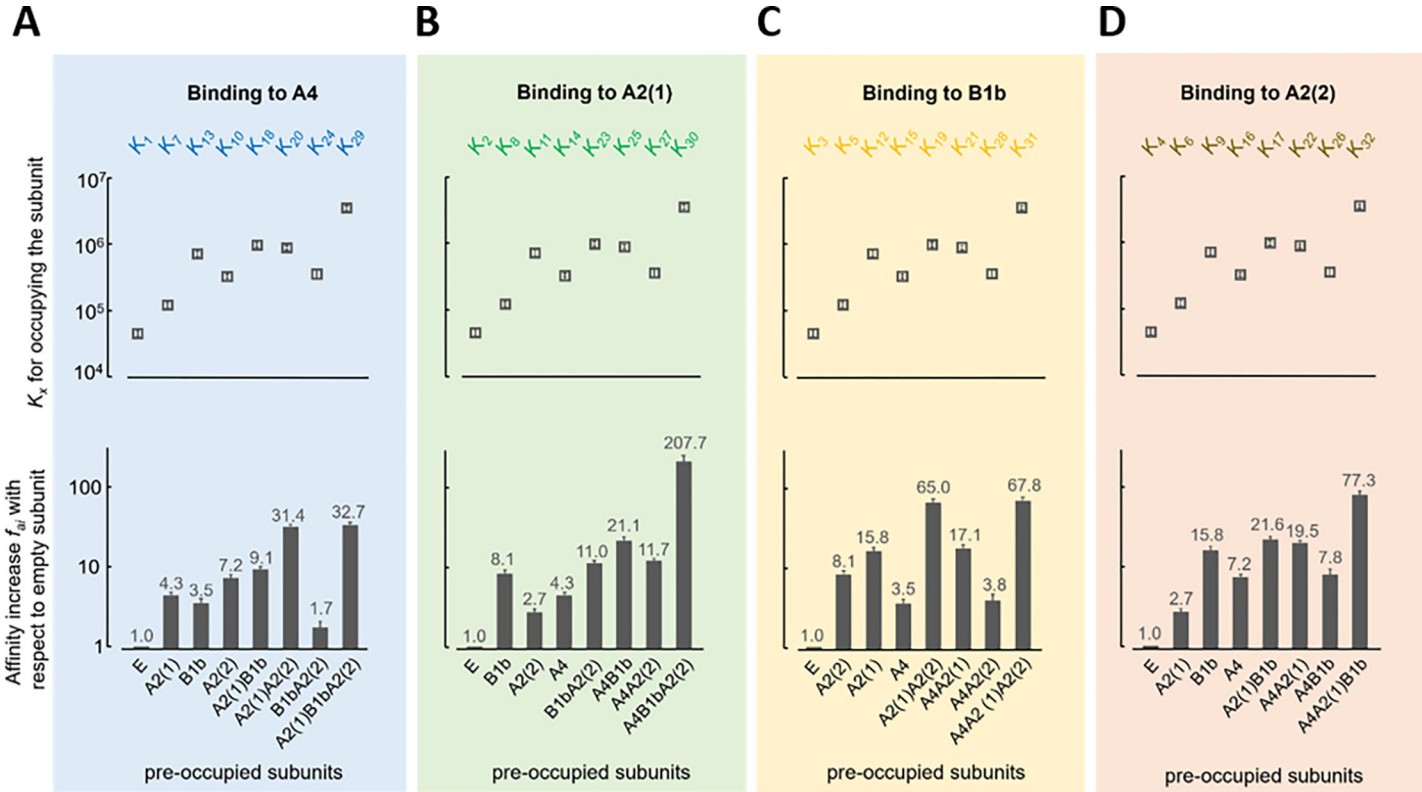

**Fig 3. Effects of pre-occupied subunits on ligand binding.** (A-D) Equilibrium association constants for the closed channel, $K_x$, and relative factors $f_{ai}$, specifying how many times the affinity of a subunit is increased by pre-occupation of the other subunits with respect to the same subunit in the empty channel, for A4, A2(2), B1b, and A2(1). 'E' indicates the empty channel. The values correspond to S6 Table.

$R$ and $T$ are the molar gas constant and the temperature in K. Thus, the four subunit-specific promotion factors $fp_n$ are translated to free energies, $\Delta\Delta G_{fp_n}$, which specify the energy contributions of the individual subunits to channel opening. These energy contributions are termed in the following 'subunit promotion energies for channel opening'. They add in a combinational way to $\Delta\Delta G_{E_0}$. For the two voltages of +100 mV and -100 mV, this results in two sets of 16 $\Delta\Delta G_{Ex}$ built by $\Delta\Delta G_{E_{0+}}$ and $\Delta\Delta G_{fp_{n+}}$ as well as $\Delta\Delta G_{E_{0-}}$ and $\Delta\Delta G_{fp_{n-}}$, respectively. Their absolute values follow the order A4>B1b>A2(1)≅A2(2) at both voltages (S8 Table).

Overall, progressive liganding forces the $E_x$ increasingly to the respective open state, some stronger at +100 mV than at -100 mV. The additivity of the $\Delta\Delta G_{fp_n}$ values is illustrated in Fig 4. At equal degree of liganding the different $\Delta\Delta G_{fp_n}$ generate notable differences in $\Delta\Delta G_{Ex}$.

## Energetic landscape of activation

Knowing all constants of the HA model for wild-type channels allowed us to consider the complete energetic landscape of the activation process.

Starting with the closed channel, we translated the $f_{ai}$ values to Gibbs free energies, $\Delta\Delta G_{Kx}$, using Eq (15), and computed the free energies of C1-C15 with respect to C0, $\Delta\Delta G_{Cx}$, (Eq 17). At $P_o$ = 0.1 (0.286 µM cGMP) there is a strong endergonic situation (grey bars in Fig 5 and S9 Table), i.e. binding is unlikely to the subunits of all states, despite the differences among them. At $P_o$ = 0.5 (1.05 µM cGMP), binding to the subunits is still endergonic, though less than at $P_o$ = 0.1. The tendency is continued at $P_o$ = 0.9 (3.85 µM cGMP) where many $\Delta\Delta G_{Cx}$ values approximate zero. Only the quadruple liganded closed channel has a slightly exergonic level. Together, the energetic differences among the states at an equal degree of liganding superimpose with the dominating effects of the ligand concentration, shifting the equilibrium to less endergonic conditions.

Next consider the free energies of the states O1-O15 in the open channel with respect to O0, $\Delta\Delta G_{Ox}$, which is possible because the respective free energies of the 32 binding constants, $\Delta\Delta G_{K_x^*}$, are given by the $\Delta\Delta G_{K_x}$ and the two adjacent $\Delta\Delta G_{E_y}$ (Eqs 18 and 19). Herein, only the results at +100 mV are considered because the effects of voltage are second-tier.

At $P_o$ = 0.1 there is still an endergonic situation although less endergonic than for the closed channel (Fig 5 and S9 Table). At $P_o$ = 0.5, the performance changes significantly: For single-, double-, and triple-liganded channels, $\Delta\Delta G_{Ox}$ values change from slightly endergonic to slightly exergonic, again superimposed by differences among the individual states. $\Delta\Delta G_{O15}$ for the quadruple-liganded channel is strongly exergonic. At $P_o$ = 0.9, the exergonic situation is further enhanced. Only the single-liganded channel generates $\Delta\Delta G_{Ox}$ around zero.

## Reliability of the parameters

To further consolidate the reliability of the exceptionally complex global fit, we used scaled unitary (SU) start vectors, containing identical elements, and varied them stochastically. If sufficiently many successful fits converging to the same absolute minimum can be identified, any influence arising from specific start vectors could be excluded. We used SU start vectors between $10^{-6}$ and $10^0$, varied each of the 29 elements stochastically over four orders of magnitude, and repeated each fit 500 times (see Materials and Methods). In total, we obtained 126 successful fits with the criterion that no parameter is negative. Subsequently, we calculated from the $\chi_x^2$ of each fit $x$ the related quantity, $C(\chi_x^2)$, using Eq (12). Setting a threshold at $C(\chi_x^2)$ = $10^{-7}$ allowed us to easily separate 114 successful fits of high consistency with $C(\chi_x^2)$ at the numerical resolution limit and to separate them from 12 fits with much larger minima, which were also inconsistent among each other (Fig 6A and 6B). The parameters of the 114 consistent

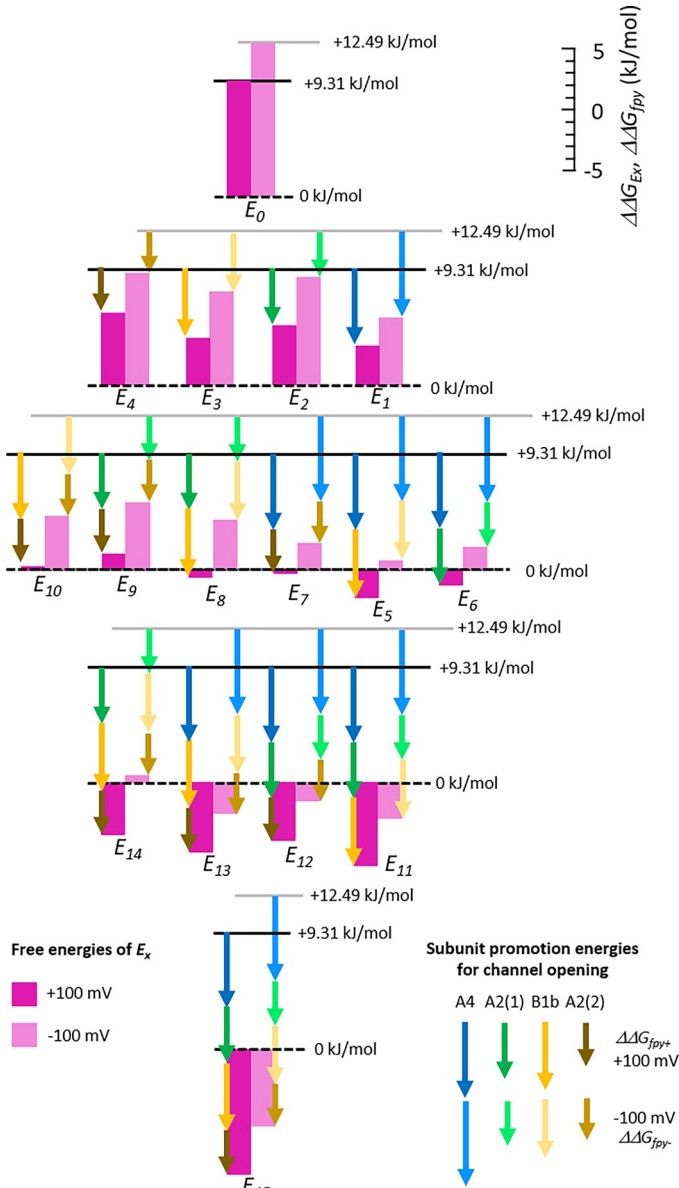

**Fig 4. Components of free energies for 16 closed-open isomerizations.** In the fit with the [29]HACO model, subunit-specific promotion factors for opening, $fp_n$, were used by which the closed-open isomerization constant of the empty channel, $E_0$, is shifted to the open state. The $fp_n$ values are either unity for the empty subunit or adopt a specific value if the subunit is liganded. Disassembling of the 16 $E_x$ leads to 16 combinational products of the five parameters at both +100 mV ($E_{0+}, fp_{1+} \ldots fp_{4+}$) and -100 mV ($E_{0-}, fp_{1-} \ldots fp_{4-}$). From these values specific subunit promotion energies for channel opening were obtained as Gibbs free energies, $\Delta\Delta G_{fp_{y+}}$ and $\Delta\Delta G_{fp_{y-}}$ (colored arrows) by using Eq (16) which were subtracted from the $\Delta\Delta G_{E_{0+}}$ and $\Delta\Delta G_{E_{0-}}$ values (magenta bars), respectively, illustrating their additive genesis. Overall, successive liganding forces the open-closed isomerizations increasingly to an open state, some more at +100 mV than at -100 mV. Hence, the different $\Delta\Delta G_{Ex}$ values at equal degree of liganding can be reduced to the energetic contributions of the different subunits.

fits match those determined above (Table 1) perfectly and are highly consistent among the fits, as indicated by standard deviations below $10^{-3}$ (S10 Table). Hence, the obtained parameters are considered to be well determined.

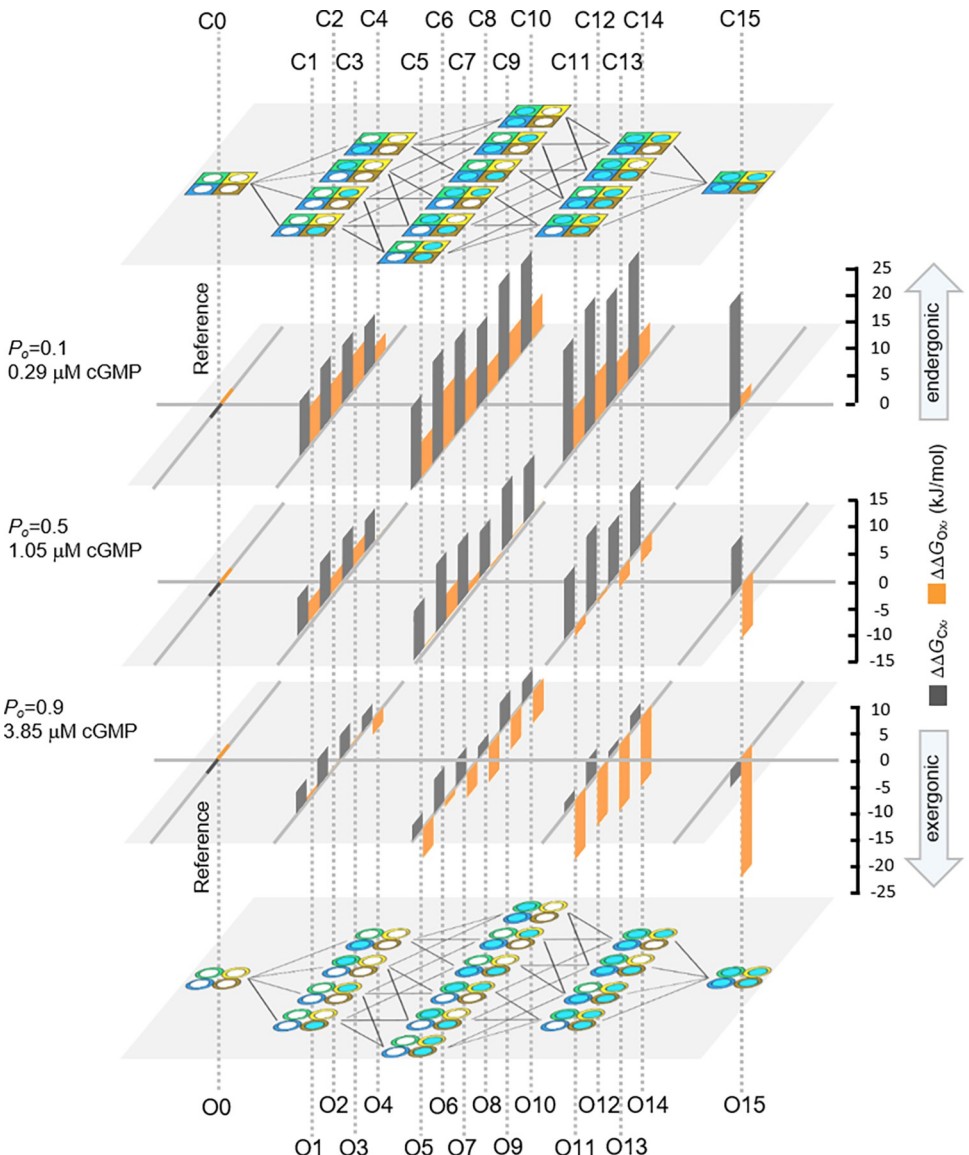

**Fig 5. Energetic landscape of the activation process.** Scheme with Gibbs free energies of the 16 closed (C0-C15) and 16 open (O0-O15) states and the HA model, as shown by the cartoons on the top and bottom, respectively. The bar graphs in-between provide the free energies $\Delta\Delta G_{Cx}$ and $\Delta\Delta G_{Ox}$ at three ligand concentrations, generating $P_o$ of 0.1, 0.5, and 0.9, respectively. The grey bars indicate the closed channel, the orange bars the open channel. In both cases the empty channel (C0, O0) was used as reference. The values and their errors are listed in S9 Table. Overall, an increasing ligand concentration shifts the diverse $E_x$ values towards the open states while the subunit-specific energetic differences at equal degree of liganding superimpose.

## Discussion

We describe for the activation of heterotetrameric olfactory channels specific subunit promotion energies for channel opening, $\Delta\Delta G_{fp_n}$, for each subunit in the context of the 32 highly cooperative binding events. The combinations of these four $\Delta\Delta G_{fp_n}$ specify together with the energy of the empty channel all 16 closed-open isomerizations of the [29]HACO model. For differentiating the 16 closed-open isomerizations at each combination of liganding we exploited a weak voltage dependence of the CARs.

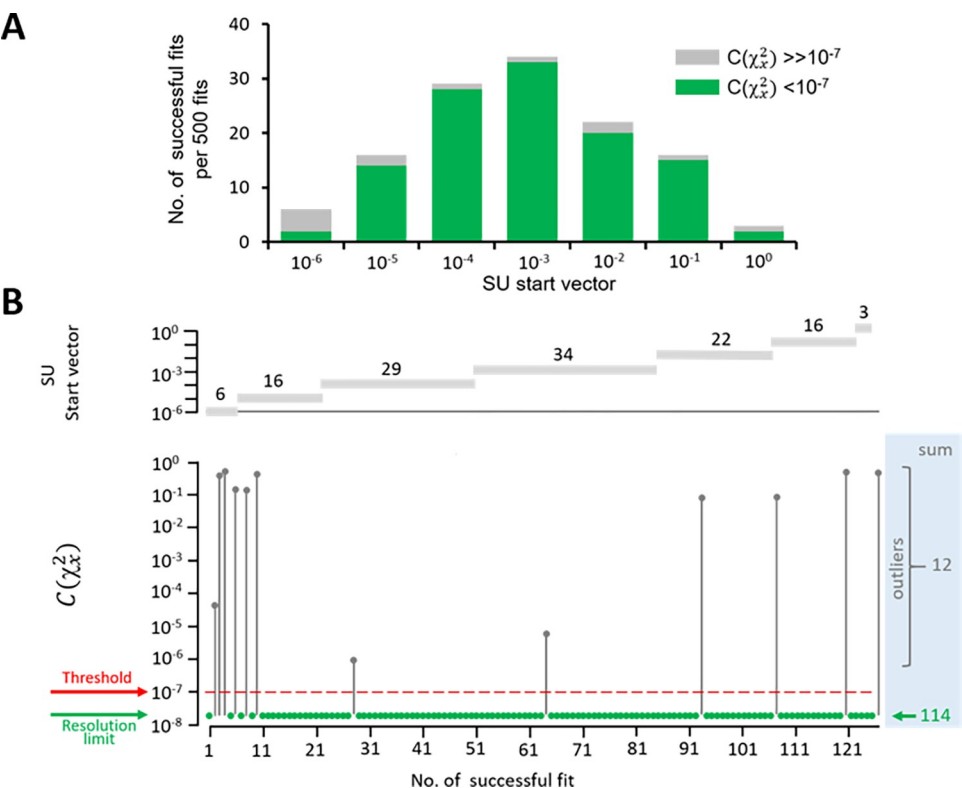

**Fig 6. Identification of successful fits with SU start vectors.** As start vectors scaled unitary (SU) start vectors, containing identical elements, were used and varied stochastically. The SU start vectors had values between $10^{-6}$ and $10^0$. The 29 elements of the SU start vectors were varied with the stochastic factor $A = 100^R$. R is a linearly distributed random number between -1 and +1. With each SU start vector 500 fits were performed, resulting in total in 3,500 fits (see Materials and Methods). Non-successful fits containing negative parameters were immediately eliminated, leaving 126 successful fits for all SU start vectors. (**A**) Histogram of the 126 successful fits as function of the value of the SU start vector. With SU start vectors of $10^{-3}$, most successful and consistent fits were obtained. The two categories were calculated as described in B. (**B**) Procedure to identify highly consistent fits at the actual minimum of $\chi^2$. From $\chi_x^2$ of each successful fit x we calculated the related quantity $C(\chi_x^2)$ by Eq (12) (see Materials and Methods). 114 successful fits with high consistency at the resolution limit could be easily separated from 12 much less consistent outliers by setting a threshold to $10^{-7}$.

Methodologically, our analysis was based on a combination of 16 concatameric channels with the sequence N-A4-A2(1)-B1b-A2(2)-C and systematically disabled, but still functional, binding sites and on a global mathematical fit with 32 intimately coupled state models.

The trustability of our global fit was demonstrated by a stochastic approach. Additional support comes from the correlation matrix of the 29 fit parameters, yielding for the vast majority of elements values near zero (S7 Fig), suggesting that the constraints in our fit are strong. Moreover, although the number of 29 free parameters seems to be enormously high, it is indeed low when comparing it with 124 parameters required when fitting Hill or double Hill functions to the 32 CARs separately (S11 Table).

As in any case of interpretation of data by a model, also our interpretations depend on the specificities of the used $^{29}$HACO model. We like to state again that the $^{29}$HACO model seems to us highly plausible for a heterotetrameric channel with four binding sites despite its considerable complexity. The key assumptions of this model are:

1. We assigned the moderate effects of voltage exclusively to the closed-open isomerizations ($E_x$) of the channel at the different combinations of liganding. This assumption is highly

plausible because the closed-open isomerizations proceed in the channel core subjected to the transmembrane electric field but not in the binding reactions ($K_x$) proceeding in the CNBDs located in the cytosol, and thus outside the transmembrane electric field.

2. For our HA models, underlying the complex [29]HACO model, we assumed that the four subunits in the channel exert only a binary closed-open action in a highly cooperative way, leading to a single opening step in our models. This assumption was supported by the observation, that the ligand concentration controls solely the open probability, $P_o$, but not the single-channel conductance (S8–S10 Figs). We therefore did not consider models of the Koshland-Nemethy-Filmer (KNF) type [36,37].

3. Our approach assumed that $\Delta\Delta G_{fp_n}$ of each occupied subunit is independent of the occupancy of the other subunits. Thus, we reduced the complex interaction of the subunits in the closed-open isomerizations into four independent functional modules. This assumption is *a priori* not without alternative. The success of our fit, however, suggested to us that this assumption is reasonable.

4. Modular functionality within the protein was also assumed for the disabling factors $fd_x$, i.e. they were taken as independent of the occupation of the other subunits [31]. Again, the success of our fit suggested to us that this assumption is reasonable.

Under these model assumptions, the phenomenon of subunit cooperativity would be confined to the level of the CNBDs. This interpretation gets nice support from recent computational studies on the allosteric signaling within the CNBD and the C-linker of structurally related HCN2 channels [38]. The authors applied MD simulations and a rigidity-theory-based approach and identified complex interactions to mediate the cAMP effect. They identified in the CNBD and the C-linker two intersubunit pathways and one intrasubunit pathway. Our analysis leaves open which structures further process the information from the CNBD and the C-linker to the pore but suggests that that this process runs independent in each subunit, i.e. there is no major additional cooperative interaction of the subunits at the level of the channel core.

The overall view is the following: Ligand binding to the different subunits of a heterotetrameric olfactory channel appears with different affinity, independent of the state of the other subunits but highly sensitive to mutagenesis (Fig 7). In contrast, subsequent conformational

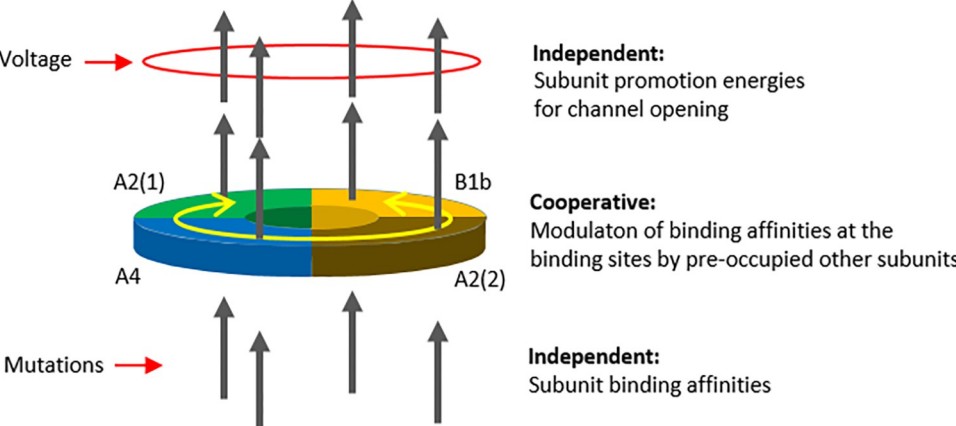

**Fig 7. Schematic summarizing the nature of the processes following ligand binding.** Pre-occupied subunits modulate the binding affinity of the other binding sites, presumably at the level of the CNBD, generating pronounced cooperativity. In contrast, both ligand binding itself and the subunit promotion energies for channel opening, $\Delta\Delta G_{fp_n}$, determined herein are independent processes.

changes in the tetrameric CNBD modulate the affinity of the CNBDs with high specificity in a cooperative fashion whereas the specific $\Delta\Delta G_{f_{p_n}}$ of a subunit, controlling the channel pore, is independent again.

## Materials and methods

### Ethics statement

The procedures had the approval of the authorized animal ethical committee of the Friedrich Schiller University Jena. The methods were carried out in accordance with the approved guidelines.

### Molecular biology and heterologous expression

The heterotetrameric concatamers assembling to olfactory CNG channels were obtained by joining the coding sequences of two CNGA2 (accession No. AF126808) subunits, one CNGA4 (accession No. U12623) and one CNGB1b (accession No. AF068572) subunit from the rat. The methods are essentially the same as reported previously [31]. The point mutations R538E (CNGA2), R430E (CNGA4) and R657E (CNGB1b) were introduced via the overlapping PCR technique, yielding all 16 combinations in the A4-A2-B1b-A4 concatamer.

Oocytes were harvested surgically under anesthesia (0.3% 3-aminobenzoic acid ethyl ester) from adult females of *Xenopus laevis*[35]. Oocytes were digested with collagenase A (3 mg/ml, Roche, Grenzach-Wyhlen, Germany) for 105 min in $Ca^{2+}$-free Barth´s solution containing (in mM) 82.5 NaCl, 2 KCl, 1 $MgCl_2$, 5 HEPES, pH 7.5. Oocytes of stage IV and V were manually dissected and injected with ~50 ng of cRNA encoding the respective channels. The oocytes were cultured at 18°C for 1 to 3 days in Barth's solution containing (in mM) 84 NaCl, 1 KCl, 2.4 $NaHCO_3$, 0.82 $MgSO_4$, 0.41 $CaCl_2$, 0.33 $Ca(NO3)_2$, 7.5 TRIS, Cefuroxim, Penicillin/Strep-tomycin, pH 7.4.

### Electrophysiology

For obtaining concentration-activation relationships at equilibrium, ensemble (macroscopic) currents, generated by hundreds to several thousands of channels, were recorded from inside-out patches with the patch-clamp technique [39]. The amplitude of the late current at +100 mV was evaluated. The patch pipettes were prepared from quartz tubing (VITROCOM, New Jersey, USA) on a P-2000 puller (Sutter Instrument, Novato, USA). The outer and inner diameter of the tubing was 1.0 and 0.7 mm. The pipette resistance was 0.5–1.7 MΩ. Both bath and pipette solution contained (in mM): 150 KCl, 1 EGTA, 10 Hepes (pH 7.4 with KOH). Recording was performed with either an Axopatch 200B amplifier (Axon Instruments, Foster City, CA), controlled by the ISO2 hard- and software (MFK, Niedernhausen, Germany), or with an EPC10 amplifier and the Patchmaster software (HEKA GmbH, Lambrecht, Germany). The sampling rate was either 2 or 5 kHz and the internal filter of the amplifier was set to either 1 or 2 kHz. The data at +100 mV were the same as published previously [31].

For single-channel recording, the patch pipettes were also fabricated from quartz tubing. The outer and inner diameter was 1.0 and 0.5 mm, respectively (VITROCOM, New Jersey, USA). The pipette resistance was 5.0–12.0 MΩ. The pipette solution contained (in mM): 150 KCl, 1 EGTA, 5 Hepes (pH 7.4 with KOH). The recording voltage was +100 mV. The data were recorded at +100 mV or 100 mV in the inside-out patch-configuration at different cGMP concentrations. Recordings from wt heteromers were sampled at 20 kHz and filtered to 5 kHz on-line. All other recordings were sampled at 40 kHz and filtered to 10 kHz on-line. For display the data were off-line filtered to 1 kHz by a Gaussian filter. Amplitude histograms were built from 10-second intervals and fitted with the sum of two normalized Gaussian functions

from which the open probability, $P_o$, and the amplitude of the unitary current, $i$, were obtained.

### Data analysis

**The global fit.** In our recent analysis of the Heteromeric Allosteric (HA) model [31], we used two equal A2 subunits and an equal opening constant $E_x$ at equal degree of liganding. Herein these two limitations were overcome by extending the HA model to the more general Heteromeric Allosteric Combinational Opening (HACO) model.

Similar to the HA model, the 16 closed states Cijkl are arranged as corners of a 4D hyper-cube (Fig 1E) [31]. Each state is assigned a four-dimensional vector with the binary indices i, j, k, and l that are either 0 or 1.

The 32 edges of the hypercube correspond to the 32 possible equilibrium association constants, $K_i$ ($i = 1, \ldots, 32$), between neighboring closed states.

For computations we adopted the assumption of microscopic reversibility [40], resulting for a 16-state model in 15 independent $K_i$ while the others are given by cycles. With respect to state C0000, this results immediately in 15 virtual equilibrium association constants $Z_i$ ($i = 1, \ldots, 15$) (violet lines in S1 Fig), defining the 32 $K_i$ by respective ratios as provided by S2 Table. Microscopic reversibility holds automatically.

Now the variable $p_c0000$ specifies the equilibrium occupation probability of the empty closed state C0000. Then the occupation probability $p_cijkl$ of each other closed state Cijkl can be easily determined from the $Zijkl$ according to

$$p_c ijkl = p_c 0000 \cdot Zijkl \cdot fd_1^a \cdot fd_2^b \cdot fd_3^c \cdot fd_4^d \cdot L^{i+j+k+l} \tag{3}$$

The ligand concentration $L$ appears as the power of the sum of the indices, reaching a maximum value of 4. The factors of disabling by mutation $fd_u$ ($u = 1, \ldots, 4$) of the four subunits have the exponents $a$, $b$, $c$, $d$ (equal to 0 or 1). An exponent is 1 if the subunit $u$ is mutated and has bound a ligand. Otherwise, the exponent is zero.

In contrast to the HA model in [31], the influence of each subunit on the opening process is treated separately. The closed-open isomerization constant of the empty channel is $E_0$. When a ligand binds to subunit $u$, an additional factor $fp_u$ ($u = 1 \ldots 4$) promotes opening. Then the total opening constant $Eijkl$ between the states Cijkl and Oijkl is given by

$$Eijkl = E_0 \cdot fp_1^i \cdot fp_2^j \cdot fp_3^k \cdot fp_4^l \tag{4}$$

This results in the probability $p_oijkl$ of the open state Oijkl to be occupied of

$$p_o ijkl = p_c 0000 \cdot E_0 \cdot fp_1^i \cdot fp_2^j \cdot fp_3^k \cdot fp_4^l \cdot Zijkl \cdot fd_1^a \cdot fd_2^b \cdot fd_3^c \cdot fd_4^d \cdot L^{i+j+k+l} \tag{5}$$

The open probability $P_o(L)$ of the whole channel is then given by

$$P_o(L) = \sum_{i=0}^1 \sum_{j=0}^1 \sum_{k=0}^1 \sum_{l=0}^1 p_o ijkl \tag{6}$$

Because the occupation probabilities of all states add up to one,

$$1 = \sum_{i=0}^1 \sum_{j=0}^1 \sum_{k=0}^1 \sum_{l=0}^1 (p_c ijkl + p_o ijkl) \tag{7}$$

this allows to eliminate $p_c0000$.

For an effective notation, we define the five subtotals $S_\alpha$ and $T_\alpha$ ($\alpha = 1, \ldots, 4$)

$$S_\alpha = \sum_{i+j+k+l=\alpha} (Zijkl \cdot fd_1^a \cdot fd_2^b \cdot fd_3^c \cdot fd_4^d) \tag{8}$$

$$T_\alpha = E_0 \sum_{i+j+k+l=\alpha} (fp_1^i \cdot fp_2^j \cdot fp_3^k \cdot fp_4^l \cdot Zijkl \cdot fd_1^a \cdot fd_2^b \cdot fd_3^c \cdot fd_4^d) \tag{9}$$

Then the open probability $P_o\,(L)$ of the whole channel is given by

$$P_o(L) = \left(\sum_{\alpha=0}^{4} T_\alpha L^\alpha\right) \cdot \left(\sum_{\alpha=0}^{4} (T_\alpha + S_\alpha) L^\alpha\right)^{-1} \tag{10}$$

The promotion factors $fp_x$ of each binding site are model parameters and their fit result delivers an insight into the contribution of each binding site to the opening process.

The squared differences between the calculated and the measured open probabilities $P_{oc}\,(L)$ and $P_{om}\,(L)$ are summed up to $\chi^2$.

$$\chi^2 = \sum_{k=1}^{n_c} \sum_{i=1}^{n_k} \frac{(P_{om}(L_{k,i}) - P_{oc}(L_{k,i}))^2}{\sigma_{k,i}^2} \tag{11}$$

The summation covers all $n_c = 16$ concatamers, at the respective $n_k$ concentrations. The weighting factors are the reciprocal of the empirical variances $\sigma_{k,i}^2$ of the mean, which have been estimated from measurements in 6 to 18 patches. Minimization of $\chi^2$ was performed with the Levenberg-Marquardt algorithm [41], providing the fit parameters and their covariance matrix $cov_{pj}$.

## Test for fit validity by stochastic variation of the start parameters

To evaluate the validity of our global fit approach with the $^{29}$HACO model minimizing $\chi^2$, we designed a strategy to avoid any biasing influence of specific start vectors. We used scaled unitary (SU) start vectors, containing identical elements, and varied them stochastically. If sufficiently many successful fits can be obtained and the non-successful fits can be identified, a major source of bias would be removed. We varied SU start vectors between $10^{-6}$ and $10^0$ in decade steps and varied each of the 29 parameters stochastically by a linear stochastic factor $A = 100^R$ where R is a random number between -1 and +1. This results in a variation of all parameters over a range of $10^4$. For each SU start vector, the fit was repeated 500 times, resulting in total in 3,500 fits. The maximum number of iterations was set to 200.

In a first step, non-successful fits were identified in which at least one parameter ran into a negative value. This criterion was chosen because a negative equilibrium constant is physically nonsense. The number of the remaining 126 successful fits, containing only positive parameters, depended on the SU start vector reaching a maximum at $10^{-3}$ (Fig 6A).

In a second step, we considered the distribution of the minima among the successful fits.

To this end, we used the $\chi_x^2$ value of each successful fit $x$ and calculated a related quantity of $\chi_x^2$ according to

$$C(\chi_x^2) = (\chi_x^2 / min(\chi_1^2, \ldots, \chi_{126}^2)) - 1 \tag{12}$$

$C(\chi_x^2)$ is zero at the minimum $\chi_x^2$ and increases at already very subtle changes. The major advantage of using $C(\chi_x^2)$ is that it allows to bring very heterogeneous $\chi_x^2$ values onto a single logarithmic scale apart from the minimum value $C(\chi_x^2) = 0$ itself. It turned out that $C(\chi_x^2)$ of 114 out of the 126 successful fits was either 0 or $2.2073 \times 10^{-8}$, the resolution limit of our calculations. If counting all fits with $C(\chi_x^2) = 0$ and $C(\chi_x^2) = 2.2073 \times 10^{-8}$ together ($C(\chi_x^2) \leq 2.2073 \times 10^{-8}$), all 126 $C(\chi_x^2)$ could be plotted (Fig 6B). The diagram illustrates the enormous consistence of $C(\chi_x^2)$ for 114 fits at the resolution limit as well as 12 outliers which were much larger and highly inconsistent. This allowed us to set a threshold at $10^{-7}$ and to distinguish the

most frequent and consistent fits at the least minimum easily from rare and inconsistent fits at much larger minima (Fig 6B). The consistent fits at the consistent least minimum are considered to provide the result of the global fit. Notably, these minima were determined without any prior knowledge of the parameters. Although these results make the identified parameters very likely to represent the best fit, these results do not finally exclude a better minimum in the 29-dimensional parameter space.

## Fits with Hill functions

For comparison, concentration-activation relationships were fitted with IgorPRO 7 (Lake Oswego, USA) by

$$I/I_{\max} = 1/(1 + (EC_{50}/[\mathrm{cGMP}])^n) \tag{13}$$

where $I$ denotes the actual current amplitude and $I_{\max}$ the maximum current amplitude at saturating cyclic nucleotide CN specified for each patch. $EC_{50}$ is the cGMP concentration evoking half maximum current and $n$ the Hill coefficient.

Part of the concentration-activation relationships required the sum of a high (H) and a low affinity (L) component

$$I/I_{\max} = A/(1 + (EC_{50,\mathrm{H}}/[\mathrm{cGMP}])^{n\mathrm{H}}) + (1-A)/(1 + (EC_{50,\mathrm{L}}/[\mathrm{cGMP}])^{n\mathrm{L}}) \tag{14}$$

The notation corresponds to that in Eq (13). $A$ is the fraction of the high affinity component.

## Computation of free energies

Free energies specifying ligand binding to a subunit with respect to the binding to this subunit in an otherwise empty channel were determined from the promotion factors $f_{ai}$ by

$$\Delta\Delta G_{Kx} = -RT\ln f_{ai} \tag{15}$$

$R$ and $T$ are the molar gas constant and the temperature in K.

The free energies for the closed open isomerizations $E_{0+}$ and $E_{0-}$, as well as for $fp_{1+}$-$fp_{4+}$ and $fp_{1-}$-$fp_{4-}$, the $\Delta\Delta G_{fp_n}$ values at -100 mV and +100 mV, were calculated accordingly by

$$\Delta\Delta G_{Kx} = -RT\ln h \tag{16}$$

where $h$ is the respective parameter.

For a given ligand concentration $L$, the free energies of all closed states C1....C15 with respect to C0 were determined by summing up the $\Delta\Delta G_{Kx}$ values, along each chosen pathway, and adding the respective free energies provided by the ligand concentration $L$, $\Delta\Delta G_L = -RT\ln(L)$, according to

$$\Delta\Delta G_{\mathrm{Cx}} = \sum_1^m \Delta\Delta G_{K_x} + m\Delta\Delta G_L (\mathrm{m} = 1\ldots4) \tag{17}$$

The free energies of the 32 binding constants, $\Delta\Delta G_{K_x^*}$, for the open channel were calculated from the respective $\Delta\Delta G_{K_x}$ and the two adjacent $\Delta\Delta G_{E_y}$ according to

$$\Delta\Delta G_{K_x^*} = \Delta\Delta G_{K_x} + \Delta\Delta G_{E_a} - \Delta\Delta G_{E_b} \tag{18}$$

where $\Delta\Delta G_{E_a}$ and $\Delta\Delta G_{E_b}$ are the free energies before and after the ligand binding, respectively. The free energy of the open states was determined in an analogue fashion to that for the closed

states

$$\Delta\Delta G_{\text{Ox}} = \sum_1^m \Delta\Delta G_{K_x^*} + m\Delta\Delta G_L (\text{m} = 1\ldots 4) \tag{19}$$

## Error propagation

The fit program provides a set of parameters together with their standard deviations and relative errors. For interpretation of the fit results we calculated from these parameters derived quantities $Y_i$ of interest. The errors of these quantities were determined according to the rules of error propagation as follows:

A quantity $Y_i$ (column vector $Y_1, \ldots, Y_m$) depending on the parameters $p_1, p_2, \ldots, p_n$ would be given by

$$Y_i = f_i(p_j)(i = 1\ldots m; j = 1\ldots n) \tag{20}$$

To obtain the statistical errors of $Y_i$, the covariance matrix of the $Y_i$, $cov_{Yi}$, is calculated from the covariance matrix of the parameters, $cov_{pj}$, with the help of the Jacobian matrix $J$

$$J = \begin{bmatrix} \dfrac{\partial Y_1(p_1,\ldots,p_n)}{\partial p_1} & \cdots & \dfrac{\partial Y_1(p_1,\ldots,p_n)}{\partial p_n} \\ \vdots & \ddots & \vdots \\ \dfrac{\partial Y_m(p_1,\ldots,p_n)}{\partial p_1} & \cdots & \dfrac{\partial Y_m(p_1,\ldots,p_n)}{\partial p_n} \end{bmatrix} \tag{21}$$

according to

$$cov_{Yi} = J \cdot cov_{pj} \cdot J^T \tag{22}$$

using the rules of matrix multiplication.

The elements $J_{i,j}$ of the Jacobian matrix are the partial derivatives of $Y_i$ with respect to the parameters $p_j$.

$$J_{i,j} = \frac{\partial Y_i}{\partial p_j} \tag{23}$$

The covariance matrix $cov_{pj}$ is quadratic of type ($n$ x $n$). The Jacobian matrix $J$ is of type ($m$ x $n$) and the covariance matrix $cov_{Yi}$ is quadratic of type ($m$ x $m$).

The standard deviations $\sigma_{i,i}$ of the $Y_i$ are obtained from the main diagonal elements of the covariance matrix $cov_{Yi}$ according to

$$\sigma_{i,i} = \sqrt{cov_{Yi}(i,i)} \tag{24}$$

In this way we calculated the errors of the association constants $K_1\ldots K_{32}$ for wild-type and mutated subunits as well as the errors of the opening constants $E_0\ldots E_{15}$ for the HACO model.

The error of the change of the Gibbs free energy, ΔΔΔG, results from the error of the factor h (Eq 16) according to

$$\Delta\Delta\Delta G = -RT\ \Delta h/h \tag{25}$$

following the rules of the differentiation of a logarithmic function. Δh was calculated from the error of the fit parameters using Eq (22) and an appropriate Jacobian matrix (21).

## Determination of $P_o$ in CARs

The equilibrium open probability, $P_o$, used in the fits was determined from the amplitude of late currents at the end of the pulses to +100 or -100 mV (Figs 2A and 2B). At +100 mV and infinite cGMP, we assumed that all constructs generate a maximum open probability of 0.99, which is based on single-channel recordings of concatenated wild-type channels [31]. Example single-channel recordings of wt concatamers, triple-mutated concatamers, quadruple-mutated concatamers as well as non-concatenated channels are shown in S8 to S10 Figs, respectively. The currents of triple and quadruple mutated concatamers, not reaching saturating currents at +100 mV, were re-scaled by $P_o$ determined in corresponding single-channel measurements for +100 mV as described [31]. At -100 mV, $P_o$ was calculated from 5 to 15 patches per construct by relating for each concatamer the late current at -100 mV to that at +100 mV, scaled by a correction factor for the different single-channel currents at both voltages and assuming also at -100 mV $P_o$ = 0.99 at infinite cGMP. The latter was based on single-channel recordings at -100 mV where at 100 μM cGMP, the non-mutated concatamer 1234 (S8 Fig) generated consistently larger $P_o$ than 0.95. The correction factor was determined using the constructs 1234m, 12m34 and 12m34m, which clearly reached saturation. Here, the ration (-current (-100mV)/current(+100mV)) was on average(n = 26) 1.10+/-0.01. This value was validated by 2 independent approaches, delivering comparable values: Un-normalized currents from an individual patch of 4m2m12m were fitted with a Hill function for -100 and +100 mV. The respective maxima and $EC_{50}$ were free, the Hill parameter was linked between the fits. The ratio of the fitted amplitudes were 1.09+/-0.06. The ratio of single channel amplitudes in a prolonged measurement for 4m2m1m2 was 1.072+/-0.005. In three concatamers (1234, 12m34, 12m34m), $P_o$ at -100 mV determined in this way slightly exceeded unity. These values were arbitrarily set to 0.99.

The errors for the parameters and constants are given as relative standard deviation (SD) in percent (rel. SD (%)). For Gibbs free energies the errors are given as SD.

## Supporting information

**S1 Fig. Virtual equilibrium association constants in the 4D hypercube.**
(DOCX)

**S2 Fig. HA models for four concatamers containing one disabled binding site.**
(DOCX)

**S3 Fig. HA models for six concatamers containing two disabled binding sites.**
(DOCX)

**S4 Fig. HA models for four concatamers containing three disabled binding sites.**
(DOCX)

**S5 Fig. HA model for the concatamer containing four disabled binding sites.**
(DOCX)

**S6 Fig. Comparison of errors obtained by fits with different models.**
(DOCX)

**S7 Fig. Color-coded matrix of correlation coefficients for the fit with the [29]HACO model.**
(DOCX)

**S8 Fig. Single-channel activity in wt concatamers N-A4-A2-B1b-A2-C (1234).**
(DOCX)

**S9 Fig. Single-channel activity in the concatamers 1m2m3m4 and 1m2m3m4m.**
(DOCX)

**S10 Fig. Single-channel activity in non-concatenated wt channels.**
(DOCX)

**S1 Table. Fit parameters of the global fit with the [17]HA model.**
(DOCX)

**S2 Table. Equilibrium constants derived from the global fit with the [17]HA model.**
(DOCX)

**S3 Table. Fit parameters of the global fit with the [24]HACO model at +100 mV OR -100 mV.**
(DOCX)

**S4 Table. Equilibrium constants derived from the global fit with the [24]HACO model at +100 mV OR -100 mV.**
(DOCX)

**S5 Table. Equilibrium constants derived from the global fit with the [29]HACO model at +100 mV AND -100 mV.**
(DOCX)

**S6 Table. Influence of pre-occupation on ligand binding for global fit with the [29]HACO model at +100 mV AND -100 mV.**
(DOCX)

**S7 Table. Closed-open isomerization constants given by the global fit with the [29]HACO model at +100 mV AND -100 mV.**
(DOCX)

**S8 Table. Subunit promotion energies for channel opening given by the global fit with the [29]HACO model at +100 mV AND -100 mV.**
(DOCX)

**S9 Table. Free energies of closed and open states obtained by globally fitting the [29]HACO model +100 mV AND -100 mV.**
(DOCX)

**S10 Table. Parameters determined with stochastically varied SU start vectors.**
(DOCX)

**S11 Table. Hill parameters for the 32 CARs at +100 mV and -100 mV.**
(DOCX)

## Acknowledgments

We thank G. Ditze, G. Sammler, F. Horn, M. Händel, K. Schoknecht, S. Bernhardt, C. Ranke and A. Kolchmeier for excellent technical assistance.

## Author Contributions

**Conceptualization:** Klaus Benndorf.

**Data curation:** Jana Schirmeyer, Thomas Eick, Christian Sattler, Ralf Schmauder, Klaus Benndorf.

**Formal analysis:** Jana Schirmeyer, Thomas Eick, Eckhard Schulz, Ralf Schmauder.

**Funding acquisition:** Klaus Benndorf.

**Investigation:** Jana Schirmeyer, Thomas Eick, Eckhard Schulz, Klaus Benndorf.

**Methodology:** Eckhard Schulz, Sabine Hummert, Ralf Schmauder, Klaus Benndorf.

**Project administration:** Klaus Benndorf.

**Resources:** Klaus Benndorf.

**Software:** Thomas Eick, Eckhard Schulz, Sabine Hummert.

**Supervision:** Eckhard Schulz, Klaus Benndorf.

**Validation:** Eckhard Schulz, Klaus Benndorf.

**Visualization:** Thomas Eick, Christian Sattler, Ralf Schmauder.

**Writing – original draft:** Eckhard Schulz, Klaus Benndorf.

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
