## [Decision Letter · Decision Letter 0]

31 May 2022

Dear Dr. Hospital,

Thank you very much for submitting your manuscript "Relating energetic opening momenta to ligand binding in individual subunits of heteromeric olfactory CNG channels" for consideration at PLOS Computational Biology.

As with all papers reviewed by the journal, your manuscript was reviewed by members of the editorial board and by several independent reviewers. In light of the reviews (below this email), we would like to invite the resubmission of a significantly-revised version that takes into account the reviewers' comments.

We cannot make any decision about publication until we have seen the revised manuscript and your response to the reviewers' comments. Your revised manuscript is also likely to be sent to reviewers for further evaluation.

Sincerely,

Alexander MacKerell

Associate Editor

PLOS Computational Biology

Jason Haugh

Deputy Editor

PLOS Computational Biology

Reviewer's Responses to Questions

**Comments to the Authors:**

Reviewer #1: The MS “Relating energetic opening momenta to ligand binding in individual subunits of heteromeric olfactory CNG channels” by Jana Schirmeyer et al deals with an important topics in the biophysics of olfactory CNG channels. These channels are heteromeric and are formed by two alpha CNGA2 subunits (A2) one CNGA4 (A4) and one CNGB1b. In the present MS, the authors describe the results they have obtained by using the concatemers N-A4-A2-B1b-A2-C concatemers where they have selectively replaced an Arginine in the CN binding site into a Glutamate so that cAMP cannot bind any more.

I like the scope and the experiments here described, but the MS in its present form is rather obscure and even though I am familiar with the topic I was not able to understand it properly. Therefore I suggest to have a mayor revision and here below are my suggestions:

1 - In the introduction it is useful and almost necessary to explain the motivations for building such model and which new insights and/or a better understanding are expected.

2 - there must be a paragraph clearly stating the assumptions on which the model/analysis is based

3 - In the discussion it is necessary to evaluate the conclusions gained by such modelling/computation

The above three suggestions are - in my view - compulsory and now I list some questions for the authors. Indeed they have the data to answer either in a final and definitive way or at least to provide a substantial better understanding for several important issues :

from the data presented in the MS it seems that some channel opening are observed when all or most of the CN binding sites are knocked off by the mutation R/E. Do the authors have some single channel recordings from these concatened and mutated channels? If so I will be delighted to see these recordings.

in long single channel recordings from the CNGA1 monomeric channel it is possible to observe occasionally long closures. Have the authors observed anything similar in their concatenated channels? If so is this observation compatible with the assumptions of their modelling?

In the MS there is a number of words, which are awkward with a not obvious meaning such:

The authors refer to Energetics of subunit opening momenta. The word “momenta” in science is the product of the mass times the velocity and I presume that the authors do not intend this.

The words endergonic and exergonic are often used and their meaning must be defined or – better – more conventional words must be used.

In the legend of Fig.1 it is stated:

“ Scheme with two closed and two open states (C0, C1, O0, O1) for a theoretical minimum ligand-gated receptor containing for the closed channel one binding step (K1) and two allosteric opening steps “

All this is hard to understand and very confusing and leaves the impression of a rather sloppy writing/editing, as if the MS is a preliminary draft and not a finished and well-polished MS

Reviewer #2: Understanding the complexities of heteromeric ion channel gating has been difficult and many electrophysiological studies to measure heteromeric CNG channel currents have been reported. Here, Schirmeyer and authors refine recently generated models to understand the energetic contributions of individual subunits to gating by ligand binding and voltage. This is an interesting question and a unique way to look at allostery and cooperatively in these channels.

The same concatemers from Schirmeyer et al. 2021 and the resulting patch clamp data recorded at +100 mV were used for both Benndorf et al. 2022 JGP and this current manuscript. For the computational model in this manuscript, the authors add patch clamp data for the same constructs recorded at -100 mV. They fit 32 equilibrium association constants and 16 equilibrium isomerization constants in contrast to the 32 equilibrium association constants and 4 equilibrium isomerization constants (Benndorf et al. 2022 JGP). Most of the results expound upon the additional and new fit constants or derivations thereof (free energies). Some of these fit constants are similar while others are more variable between the previous HA (+100 mV) and HACO models (+100 mV and -100 mV).

Major point

Improvement in fitting are less than 5% relative standard deviations (Figure S6) or approximately equivalent to the error of estimated constants (Table 1), which I believe reduces the enthusiasm for the actual enhancement of the HACO model (+100 mV and -100 mV) over the HA model (+100 mV). Could the authors comment on this?

Minor points

Why are equivalent KCl concentrations in the bath and pipette for the patch clamp experiments?

Show single channel activity of natural CNGs at -100 mV to complement Figure S8. It would also be nice to see representative comparisons for mutants between natural and concatemer channels at the different Po's described at both -100 mV and +100 mV.

While these concatemers were used in previous publications to generate models, are the authors concerned with the differences in kinetics between the natural CNG channel and the concatemer and what this could mean for their conclusions (Figure 2A and 2B)?

Can the authors speculate why the constants are so different between +100 mV and -100 mV?

**Have the authors made all data and (if applicable) computational code underlying the findings in their manuscript fully available?**

Reviewer #1: **No: **I have not seen neither the code nor the original data, but they could be somewhere and I did not notice

Reviewer #2: Yes

PLOS authors have the option to publish the peer review history of their article (what does this mean?). If published, this will include your full peer review and any attached files.

Reviewer #1: **Yes: **Vincent Torre

Reviewer #2: No
---

## [Decision Letter · Decision Letter 1]

11 Jul 2022

Dear Dr. Benndorf,

We are pleased to inform you that your manuscript 'Subunit promotion energies for channel opening in heterotetrameric olfactory CNG channels' has been provisionally accepted for publication in PLOS Computational Biology.

Best regards,

Alexander MacKerell

Associate Editor

PLOS Computational Biology

Jason Haugh

Deputy Editor

PLOS Computational Biology

Reviewer's Responses to Questions

**Comments to the Authors:**

Reviewer #1: the ms is acceptable in its present version

Reviewer #2: In my opinion, I believe the authors should consider that a 10% standard deviation versus a roughly 12% standard deviation between the different models is probably not that significant. While it is interesting to think about additional parameters and what this can mean in terms of gating, I still do not see a major mathematical improvement from the simpler model with fewer terms. It's perhaps even more puzzling that the improvements in the HACO model (+100 mV and -100 mV) seemingly invoke voltage criteria for channels that are thought to have very weak voltage dependence, as the authors noted. Reviewer comments were addressed adequately.

**Have the authors made all data and (if applicable) computational code underlying the findings in their manuscript fully available?**

Reviewer #1: Yes

Reviewer #2: Yes

PLOS authors have the option to publish the peer review history of their article (what does this mean?). If published, this will include your full peer review and any attached files.

Reviewer #1: **Yes: **Vincent Torre

Reviewer #2: No

---

## [Editor Report · Acceptance letter]

17 Aug 2022

PCOMPBIOL-D-22-00564R1 

Subunit promotion energies for channel opening in heterotetrameric olfactory CNG channels

Dear Dr Benndorf,

I am pleased to inform you that your manuscript has been formally accepted for publication in PLOS Computational Biology. Your manuscript is now with our production department and you will be notified of the publication date in due course.

With kind regards,

Zsofia Freund
